# Emissions and Control Assessment of Volatile Organic Compounds from a Typical Chemical Enterprise

**Lin Wang [1], Dong Lin [1], Rui Liu [2], Jing Li [1] and Xiuyan Xu [3,*]**

1   Tianjin Eco-Environmental Monitoring Center, Tianjin 300191, China
2   The Faculty of Environment and Life, Beijing University of Technology, Beijing 100124, China
3   China National Environment Monitoring Centre, Beijing 100012, China
*   Correspondence: xuxy@cnemc.cn

**Abstract:** Emissions from the chemical industry are among the important sources of atmospheric volatile organic compounds (VOCs), which face control challenges such as multiple emission sources, high emission intensity and complex pollutant types. In this study, a typical chemical enterprise is selected as the research object, and VOCs characteristics such as emission amount, concentration and composition are analyzed; end-pipe treatment of VOCs is evaluated and control suggestions are proposed. Results show that the annual emission amount of VOCs from organized stacks was 64.08 tons, accounting for 72% of total emissions. Cyclohexane and xylene were the major components during the emission. The filling process was the largest contributor in the case of unorganized emission. As far as end-pipe treatment was concerned, ultraviolet (UV) photolysis varied greatly between 12–80%, indicating unstable removal efficiency. Finally, several measures concerning whole-process control were suggested.

**Keywords:** volatile organic compounds; emission; treatment; chemical enterprise

## 1. Introduction

Volatile organic compounds (VOCs) are important precursors of tropospheric ozone and secondary organic aerosols; they are also involved in atmospheric chemical reactions [1] that could bring acid rain, photochemical smog and other environmental problems [2,3]. Studies have shown that VOCs emissions can enhance atmospheric oxidation and malodors [4,5]. In recent years, control effects for $SO_2$ (sulfur dioxide), $NO_X$ (nitrogen oxides) and particles were obvious in China. However, VOCs are not so well-controlled as $SO_2$ and $NO_X$, and the impact of VOCs on the atmosphere is increasingly prominent [6]. So, the problems of atmospheric compound pollution triggered by VOCs are the main challenges at present in China [7–9].

The sources of VOCs are divided into natural and anthropogenic sources, among which industrial emissions are the largest contributor of anthropogenic sources in China, and the chemical industry contributes significantly to industrial-source emissions [10,11]. Most chemical enterprises have a wide variety of chemicals and yield different VOCs emission components including aldehydes, ketones, alkanes and so on [12]. At present, many chemical enterprises still have many difficulties in VOCs control, such as the low removal efficiency of treatment facilities, limited application of different control technologies, insufficient attention to VOCs management and a low degree of automation [13–15]. Besides, unorganized emissions and limited monitoring make VOCs control more difficult [16–18].

In this study, a typical chemical enterprise was selected as a case study to investigate the VOCs-related processes, raw and auxiliary materials, emissions and other information, to analyze VOCs emission characteristics, to evaluate control measures and to put forward corresponding treatment measures and suggestions for better control.

## 2. Methodology

### 2.1. Sites Description and VOCs Measurement

A typical chemical enterprise (114.64, 22.76) located in the Daya Bay Petrochemical Park, Huizhou City, Guangdong Province, was selected to carry out a study on the characteristics of VOCs emissions. The main products of the enterprise include resin, ultraviolet (UV) monomer and paint products. The resin products are alkyd resin, polyethylene (PE) unsaturated resin and UV resin with an annual output of 30,000 tons, UV monomer with an annual output of 10,000 tons, and paint products with an annual output of 40,000 tons of polyurethane (PU), PE, nitrocellulose (NC) and other wood paint, solidified lacquer box waterborne wood lacquer and other series of products.

According to the purpose of this study, the research mainly consists of three parts, which are the LDAR (leak detection and repair) test, organized and unorganized emissions and end-pipe emission control assessment. The relevant monitoring of this study was completed in May 2021 for one week.

LDAR is a technology used to reduce unorganized emissions of VOCs and is now widely adopted internationally [19]. The valves, flanges, pumps, vacuum gauges, connectors and open lines on the production unit are monitored by a portable analyzer in accordance with certain methods. During testing, the sampling probe is placed on the relevant part of the equipment or device where the leakage discharge is likely to occur and moved along its periphery at a speed of less than 10 cm/s while paying attention to the instrument readings. If the instrument reading exceeds the leakage standard, it is repaired within a certain period of time according to the relevant work flow and reinspected. The above steps are repeated to reduce the leakage points of the equipment and achieve the purpose of controlling the disorganized discharge of the production plant. LDAR detection in this part was conducted in strict accordance with the relevant technical requirements of the "Technical Guidelines for the Detection of Volatile Organic Compounds Emitted from Leakage and Open Liquid Level" (HJ733–2014) [20], and the leakage testing of all types of equipment components of the production units was carried out. The instrument used in this study was a portable organic gas analyzer (TVA-1000B, Thermo, USA). Operating parameters: detection range: PID detector dynamic range: 0–2000 ppm; low detection limit: 100 ppb benzene; FID detector dynamic range: 0–50,000 ppm; low detection limit: 200 ppb benzene; response time: 2 s; sample flow rate: 1000 cc/min; signal output: 0–2 VDC. In addition, a Fourier extracted infrared spectrometer (FACE-EB3000, Mastek, Taiwan) was used for component determination of the specific leaked VOCs in this study [21,22]. After the beam from the infrared light source passes through the interferometer, the interferogram is formed on the detector, which is collected by the acquisition card and input into the computer. After Fourier transform processing, the corresponding infrared absorption spectra of the sample gas are obtained and qualitative and quantitative analyses are carried out.

The organized emission test mainly refers to the relevant regulations of HJ732–2014 [23]. This study adopted negative pressure sampling with a sampling bag in front and air pump in back. After the samples were collected by the on-site air-bag method, they were quickly transferred to the summa tank by using the quick plug connector, and then sent back to the laboratory for preliminary preconcentration treatment. The MS mass spectrum in GC-MS/FID (6890GC-5975N, Agilent, Palo Alto, California, USA) was used for the qualitative analysis of specific components. The TO-15 standard gas + PAMS standard gas were used as the standard curve, and the FID was used for quantitative analysis [24,25]. Its working parameters are as follows: 200 °C at the injection port; temperature rise of chromatographic column: 35 °C for 2 min, 5 °C/min to 150 °C and then 20 °C/min to 220 °C for 5 min; column flow: 1.2 mL/min. Specific analysis can be seen in the reference literature [26].

In addition, TVA-1000B was also used for processes related with other unorganized emissions such as storage tank breathing, production filling release, wastewater fugitive diffusion and so on [27]. As for the end-pipe emission control assessment, two TVA-1000B were also use simultaneously at the inlet and outlet of the ultraviolet (UV) photolysis device to test VOCs concentrations for about 30 min for the removal-efficiency calculation.

## 2.2. Emission Amount Calculation

The chemical industry is different from other industries in that most of the VOCs emissions are unorganized emissions [28]. The organized emissions are generally (1) the organized emissions from the tail gas of the device in the production processes (2) the flue gas generated in the heating and combustion process of hot furnace, boiler, other equipment, etc. [29]. The organized and unorganized emission amount calculation mainly refers to the simplified accounting methods for petroleum refining, petrochemical industry VOCs emissions to analyze the processes links and their possible characteristics of emission pollution and calculate the emission based on the actual monitoring situation [30].

## 3. Results and Discussion

### 3.1. VOCs Characteristics

#### 3.1.1. Organized Emission

At present, VOCs emitted from paint workshops, resin workshops, UV monomers, monomer wastewater fugitives, paint research rooms and so on are collected together and treated uniformly before entering the atmosphere. The determination of relevant VOCs components were conducted, and the concentration of total VOCs was 165 mg/m$^3$ during the monitoring period. As can be seen from Table 1, the characteristic emission pollutants of the enterprise were cyclohexane, toluene, xylene (o-, m-, and p-), ethyl acetate, butyl acetate and styrene. Emissions of these species accounted for about 80% of the total VOCs. Xylene and cyclohexane were the highest among these detected VOCs. Xylene mainly originated from the reaction reactor in the resin workshop and the dispersion and grinding in the paint workshop. Meanwhile, cyclohexane was mainly generated by the reaction in the UV monomer workshop.

**Table 1.** Information about the VOC components emitted from the stack.

| No. | Testing Items | Concentration (mg/m$^3$) |
|:---:|:---:|:---:|
| 1 | Acetone | 0.24 |
| 2 | Isopropyl alcohol | 0.28 |
| 3 | 1,3-Pentadiene | 0.45 |
| 4 | 2-Methyl-dipropanol | 0.92 |
| 5 | Trichloromethane | 5.35 |
| 6 | Methylcyclopentane | 0.67 |
| 7 | Cyclohexane | 27.05 |
| 8 | Methylcyclohexane | 0.87 |
| 9 | Benzene | 0.29 |
| 10 | Ethyl acetate | 4.50 |
| 11 | Toluene | 5.99 |
| 12 | Butyl acetate | 7.45 |
| 13 | Tetrachloroethylene | 3.79 |
| 14 | Styrene | 5.69 |
| 16 | m/p-Xylene | 19.33 |
| 17 | o-Xylene | 9.51 |
| 18 | Phenol | 1.89 |
| 19 | Phenylacetate | 1.73 |

The annual emissions of VOCs from the enterprise's organized emissions were calculated based on the detection method combined with the exhaust emissions and the annual operating hours of the enterprise (7200 h):

$$E_{emissions} = C_{NMHC} \times V_{volume} \times t = 64.080 \text{ tons} \qquad (1)$$

#### 3.1.2. Leakage of Production Units

A relatively comprehensive LDAR test was conducted on the paint, resin and monomer workshops. The results showed that there was a certain degree of leakage of units in the

three workshops, with 5880 leak points; the specific leaking species are shown in Figure 1 and the overall leakage situation is shown in Figure 2.

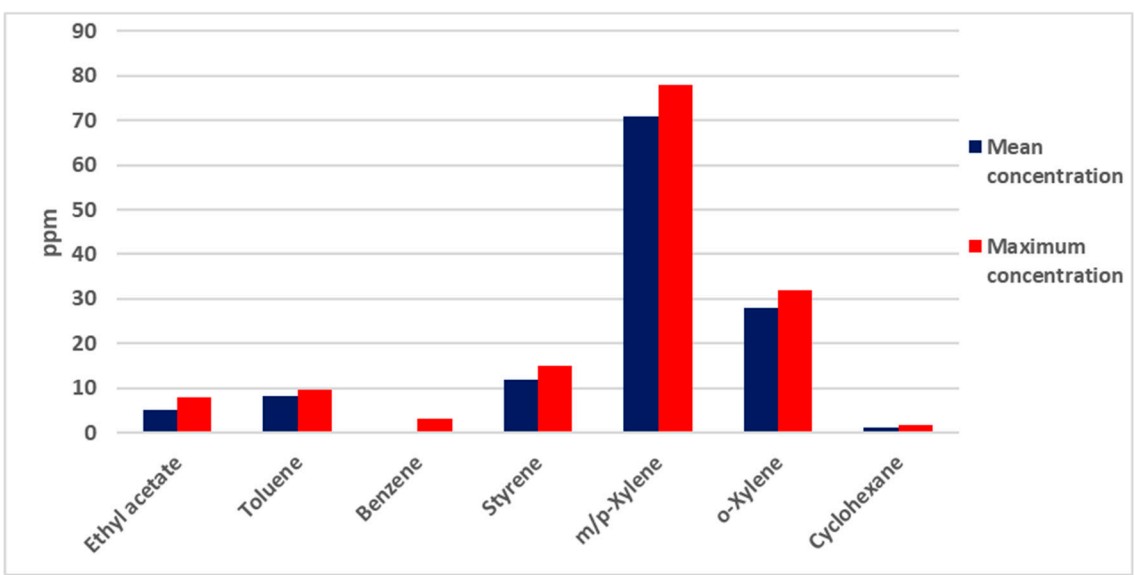

**Figure 1.** Leaking substances from production unit components.

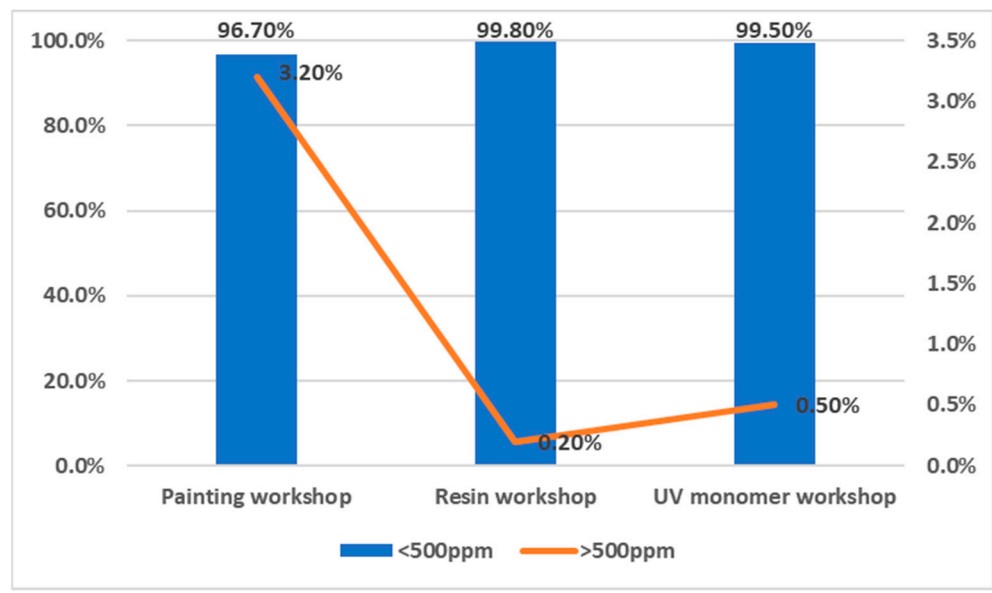

**Figure 2.** Leakage of production unit components.

The number of points with leakage > 500 ppm in the paint, resin and monomer workshops were 14, 5 and 9, respectively, which were classified as significant leakage, and the main leaking substance was xylene (m-, o-, p-). The rate correlation equation method was used to calculate the VOCs emissions from resin, monomer and paint production workshops, and their emission rates were 0.192 kg/h, 0.135 kg/h and 0.146 kg/h, respectively. The annual VOCs leakage amount of the enterprise was 3.4 tons.

### 3.1.3. Storage Tank Breathing Emissions

At present, the storage tanks of enterprises are all underground tanks, which are mainly used to store diesel, acrylic acid, cyclohexane, dicyclopentadiene, xylene, styrene, butyl acetate, toluene and other materials. The enterprise adopts the method of heavy oil absorption to deal with the relevant respiratory emissions. By using the formula in EPA

AP-42 7.1 and combining with the status of the company's existing tank filling [31], the existing storage tank emissions of the enterprise were calculated to be 5.14 tons. Among them, the storage tank containing acrylic acid, dimethyl carbonate and xylene yielded more emissions, which were 1.39 t/a, 0.85 t/a and 0.62 t/a, respectively.

### 3.1.4. Product Filling Release

The main products of the enterprise, such as paints, curing agents, and UV monomer are filled by barrels. It was found that there was a certain amount of fugitives during the filling process. After obtaining the information of its concentration, barrel-mouth wind speed, barrel filling time and barrel diameter, the fugitive amount was calculated to be 0.27 kg/h. Similarly, the fugitive amount of resin and UV filling process were calculated as 0.012 kg/h and 0.026 kg/h, respectively. According to the EIA data of the enterprise, combined with the actual application status of the enterprise, the VOCs emissions caused by filling were 1.83 kg/h. Among them, the paint workshop yielded the most, which was 1.62 kg/h. The component with the highest concentration was m-xylene, followed by xylene (m, o, p), styrene, etc. According to the accounting of VOCs produced by filling, the annual emission was 13.18 tons.

### 3.1.5. Assessment of VOC Treatments

In this study, UV photolysis oxidation was used to deal with VOCs emitted from the paint workshop, resin workshop, monomer workshop, wastewater and so on. VOCs removal efficiency were conducted at the inlet and outlet of the UV photolysis treatment. The results are shown in Figure 3.

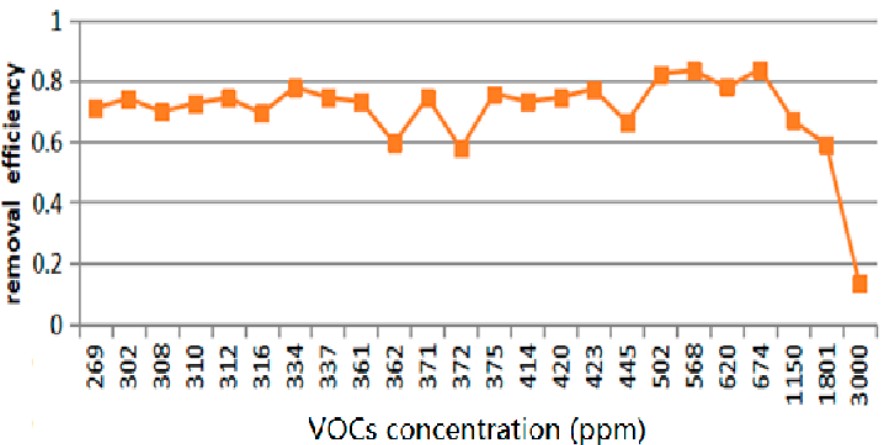

**Figure 3.** VOC treatment effect of UV photolysis device.

As can be seen from Figure 3, the emission of VOCs fluctuated greatly, especially in the 130th minute during continuous monitoring. There was a high concentration of VOCs around 2142 mg/m$^3$, which was mainly due to the intermittent production of the enterprise, and the high concentration value was mainly influenced by the production of UV monomer. By comparing the removal efficiency at different concentrations, it could be found that the removal efficiency of UV photolysis treatment facilities of this company was basically between 60−80%, but when the VOCs exceeded 714 mg/m$^3$, the removal efficiency dropped sharply to a minimum of about 12%. According to the preliminary analysis, the reason might be that the UV lamp was not powerful enough, which did not produce enough oxygen radicals to cause oxidative decomposition of VOCs. The removal efficiency varied for specific VOCs species, with a low removal efficiency for benzene and a high removal efficiency of about 85% for cyclohexane.

## 4. Conclusions

LADR monitoring results show that significant leakage points of elements from production devices account for a relatively small proportion, and VOCs emissions caused by element leakage from production devices account for a relatively small proportion. Therefore, element leakage from production devices is not the main source of pollution, and enterprises need to regularly conduct LDAR monitoring to reduce leakage. The annual emissions of organized exhaust is 64.08 tons, and its emissions account for 72% of the overall emissions. Organized emissions should become the focus of enterprise VOCs control. Enterprises should further improve pollution control equipment, or choose an activated carbon concentration + catalytic combustion mode for transformation, to improve the efficiency of waste gas treatment. As a part of the unorganized emission, the leakage generated during the product filling process deserves attention. Enterprises should improve the design and layout of gas-collecting pipelines to reduce the unorganized emissions caused by them. The VOCs leakage caused by big and small breathing of storage tank and wastewater escape processes was 5.14 tons and 0.13 tons, respectively. More stringent control measures should be considered for storage tanks and wastewater ponds. According to the analysis results, enterprises should be based on their own process characteristics and emission characteristics targeted to carry out comprehensive and multi-angle adaptive control of pollution, improve the level of enterprise equipment, implement on-site management measures, and then effectively control the emission of VOCs.

**Author Contributions:** Conceptualization, L.W.; methodology, D.L.; investigation and data curation, R.L.; writing—original draft preparation, J.L.; supervision, X.X. All authors have read and agreed to the published version of the manuscript.

**Funding:** This research received no external funding.

**Institutional Review Board Statement:** Not applicable.

**Informed Consent Statement:** Not applicable.

**Data Availability Statement:** Data is available on request.

**Conflicts of Interest:** The authors declare no conflict of interest.

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
