# Peer review of "Emissions and Control Assessment of Volatile Organic Compounds from a Typical Chemical Enterprise"

_atmosphere, doi:10.3390/atmos14020206_

Round 1

Reviewer 1 Report

In this MS, Wang et al. selected a typical chemical plant, reported its VOC emission characteristics, and made some control management suggestions. Given that VOC control is the focus and difficulty of ozone and particulate matter pollution control, this article has some scientific value for controlling industrial VOC emissions. However, this MS cannot be strictly considered as a scientific paper; it is essentially an experimental report and a not-so-qualified one at that.

1. I am particularly dissatisfied with the description of the method section.

1.1. Although I understand that it is not appropriate for the authors to reveal the exact name and location of the plant, the reader needs more information about the plant. The authors should describe in detail the sampling location, sampling duration, sampling strategy, etc., rather than simply referring readers to previous studies or standards.

1.2. We need to know the performance and key parameters of the instrument in addition to the name of the instrument.

1.3. Why not just use summa containers for sampling?

2. Line 106. In this equation, t is 7200 hours. Is the relationship between emission concentration and production linear? If it is not linear, this equation will be difficult to find the total annual emission of VOC.

3. Figure 2 conveys a very simple message, but it is very confusing. Please reconsider a better way to express it.

4. Throughout the MS, the suggestions for VOC control are only slightly mentioned in the conclusion section and are only descriptive, lacking connection to the measurement results. In other words, the findings of this paper are not consistent with the title of this paper.

Reviewer 2 Report

1. Add reference:“Technical Guidelines for the Detection of Volatile Organic Compounds Emitted from Leakage and Open Liquid 61 Level”(HJ733-2014),

2. What is the sampling time?

3. Table 1 should clearly give the specific release amount of each substance, which cannot be expressed by spectral area ratio. Explain how the various substances in the experiment are quantified.

4. Figure 1. Meaning of each axis in 3D?

5. Comparing standards to illustrate the scientific basis of various release calculations, what is the actual verification error?

6. Indicate the variance analysis for all graph measurements in the paper, indicating the number of repeated measurements? Where and how is it measured?

7. Handheld instrument measurement data flapping, how to deal with instability?

8. The corresponding analysis in Figure 2 is not clear. It is necessary to clarify the type and concentration distribution of leaked substances in different types of enterprises.

Round 2

Reviewer 1 Report

Thanks to the authors for addressing my comments. I have no other technical concerns, but obviously the English language needs to be greatly improved.

Reviewer 2 Report

  • After targeted modification, the paper basically meets the requirements.